# Relationship between Dietary Magnesium Intake and Metabolic Syndrome

**DOI:** 10.3390/nu14102013

**Published:** 2022-05-11

**Authors:** Yingying Jiao, Weiyi Li, Liusen Wang, Hongru Jiang, Shaoshunzi Wang, Xiaofang Jia, Zhihong Wang, Huijun Wang, Bing Zhang, Gangqiang Ding

**Affiliations:** 1National Institute for Nutrition and Health, Chinese Center for Disease Control and Prevention, Beijing 100050, China; jyy2227085940@163.com (Y.J.); liwy@ninh.chinacdc.cn (W.L.); wangls@ninh.chinacdc.cn (L.W.); jianghr@ninh.chinacdc.cn (H.J.); wangssz@ninh.chinacdc.cn (S.W.); jiaxf@ninh.chinacdc.cn (X.J.); wangzh@ninh.chinacdc.cn (Z.W.); wanghj@ninh.chinacdc.cn (H.W.); zhangbing@chinacdc.cn (B.Z.); 2Key Laboratory of Trace Elements and Nutrition, National Health Commission, Beijing 100050, China

**Keywords:** dietary magnesium intake, metabolic syndrome, dose-response relationship

## Abstract

Metabolic syndrome (MetS) is increasingly prevalent, and the relationship between dietary magnesium and MetS remains controversial. Therefore, we aimed to explore the association and dose-response relationship between dietary magnesium intake and MetS and its single component. The sample was adults aged 18 years and above who participated in at least two follow-up surveys in 2009, 2015 and 2018. Food consumption data were collected from three consecutive 24-h dietary recalls. The multivariate Cox proportional risk regression model and restricted cubic spline (RCS) model were used to analyze the association and dose-response relationship between dietary magnesium intake and MetS and its components. In our study, 6104 subjects were included, with a total follow-up of 37,173.36 person-years, and the incidence was 33.16%. Cox regression analysis showed that the multivariable-adjusted Hazard Ratio (HR) for MetS comparing the highest to the lowest quintile of dietary magnesium intake was 0.84 (95% confidence intervals [CI] = 0.71–0.99). Central obesity, elevated TG, elevated blood pressure and elevated blood glucose were reduced by 18%, 41%, 20% and 42%, respectively. The risk of decreased HDL-C was reduced by 23% in the third quintile of dietary magnesium intake, with a slightly increased risk in the highest group. RCS analysis showed that the overall and non-linear associations between dietary magnesium and MetS and its components were statistically significant, the risk of them decreased significantly when magnesium intake was lower than 280 mg/day, and then the curve leveled off or slightly increased.

## 1. Introduction

Metabolic syndrome (MetS) is a clinical syndrome characterized by obesity, hyperglycemia, hypertension and dyslipidemia, which seriously affects body health [1]. In recent years, the prevalence of MetS has shown an increasing trend both at home and abroad, affecting about 10−50% of adults in the world [2]. According to the nutrition and health status monitoring data of Chinese residents, the prevalence of MetS among Chinese adults was 11.0% from 2010 to 2012 [3] and 18.1% in 2015 [4]. In addition to obesity and sedentary behavior, nutrients are thought to play an important role in the development of MetS [5]. Magnesium is an essential nutrient found in green leafy vegetables, whole grains, legumes and nuts [6]. Studies have found that dietary magnesium can regulate systemic inflammation [7], change lipid abnormalities [8] and reduce the risk of hypertension and cardiovascular diseases [9,10]. The association of dietary magnesium intake with MetS and its components remains controversial [11]. Two meta-analyses showed a negative correlation between magnesium intake and MetS [12,13]; however, in individual studies, a significant association was found only in women or not observed at all [14,15], and most of the existing studies are cross-sectional studies. Given the increasing prevalence of MetS and the low dietary magnesium intake of Chinese residents [16], it is particularly important to determine whether low magnesium intake is associated with an increased risk of MetS. Therefore, this study analyzed the association between dietary magnesium and MetS and its single component using longitudinal study data from the China Health and Nutrition Survey in 2009, 2015 and 2018 and provided a reference for adjusting dietary structure to prevent MetS.

## 2. Materials and Methods

### 2.1. Study Design and Subjects

The study used data from the China Health and Nutrition Survey (CHNS), a longitudinal follow-up study initiated in 1989 that had completed 11 waves of follow-up. A stratified multistage cluster random sampling method was used in 15 provinces (autonomous regions, municipalities) of Heilongjiang, Shandong, Henan, Guangxi, Liaoning, Jiangsu, Hubei, Hunan, Guizhou, Beijing, Shanghai, Chongqing, Shaanxi, Yunnan and Zhejiang. For each wave of investigation, nutrition and health-related information was collected from the community, family and individual levels for the same population, including demographics, lifestyle, diet, economy, community conditions, etc. Refer to the relevant literature for specific investigations [17,18,19]. The collection and testing of blood samples were added in 2009, 2015 and 2018. Adults aged 18 years and above from the 2009, 2015 and 2018 surveys were selected for this study. We excluded those who had demographic information deficiency (*n* = 1292), pregnant and lactating women (*n* = 130), those who had dietary data deficiency (*n* = 2911), abnormal daily energy intake (males: >6000 kcal or < 800 kcal; Women: > 4000 kcal or < 600 kcal) and abnormal BMI (<14.0 kg/m^2^ or >45.0 kg/m^2^) (*n* = 178), those with MetS at baseline (*n* = 4648), and those only participated in the follow-up once (*n* = 6113). A total of 6104 subjects were finally included in the study (Figure 1). Analysis of a single component, independent of MetS status at baseline, identified specific study population subgroups for each single component of MetS. The Institutional Review committees of the University of North Carolina at Chapel Hill and the National Institute for Nutrition and Health, Chinese Center for Disease Control and Prevention, approved the survey (No. 201524). Participants provided their written, informed consent.

### 2.2. Evaluation Indicators

#### 2.2.1. Dietary Nutrients

In each wave of CHNS, the data of individual food consumption were collected by the three consecutive 24-h dietary recalls (two weekdays and one weekend day), and the usage of edible oil and condiments was collected by the household weighing method and allocated to individuals according to the ratio of individual energy consumption in the household. The consumption of various foods, edible oils and condiments collected was converted into nutrient intake using the “Chinese Food Composition Table” [20,21]. The average daily nutrient intake was calculated based on the person-days during the survey period. Energy, dietary fiber, dietary calcium and magnesium were included in this study.

In prospective analyses, considering that potential changes in diet after the development of the disease may confound the relationship between dietary magnesium intake and MetS and its components, updating of dietary information was stopped upon diagnosis of the disease. If the subjects entered the cohort in 2009 and developed in 2015, the dietary magnesium intake values of 2009 were used; if developed in 2018, the average dietary magnesium intake values of 2009 and 2015 were used. If subjects entered the cohort in 2015 and developed in 2018, magnesium intake values of 2015 were used. In addition, dietary magnesium intake after energy standardization was used to control the effect of total energy intake [22].

#### 2.2.2. Diagnostic Criteria for MetS

According to the joint statement of the International Diabetes Federation (IDF) in 2009 [1]: Central obesity: waist circumference ≥85 cm in men and ≥80 cm in women; Elevated triglyceride (TG): ≥1.7 mmol/L or under treatment; Decreased high-density lipoprotein cholesterol (HDL-C): <1.0 mmol/L in men and <1.3 mmol /L in women or under treatment; Elevated blood pressure: systolic blood pressure ≥130 mmHg, diastolic blood pressure ≥85 mmHg or being treated for essential hypertension; Elevated plasma glucose: fasting blood glucose (FPG) ≥5.6 mmol/L or previously diagnosed with diabetes. Having ≥3 of these risk factors was defined as MetS.

#### 2.2.3. Measurement of Indicators

Calibrate the instrument before taking physical measurements. Waist circumference, height, weight and blood pressure were measured using a non-retractable material flexible ruler, SECA206 altimeter, electronic weight scale and standard mercury sphygmomanometer (SBP and DBP were determined according to Korotkoff sound, and three consecutive standard measurements were taken for each subject to take the average value), with an accuracy of 0.1. Overnight fasting blood samples were collected by trained nurses, and an array of biochemical indexes were measured with strict quality control. Fasting plasma glucose was measured by a Roche 702 instrument and Roche reagent using hexokinase method; TG and HDL-C were measured by a Hitachi 7600 Automated Analyzer by the Laboratory of China-Japan Friendship Hospital in 2009. Fasting plasma glucose concentration was measured by glucose oxidase-phenol and amino phenazone (GOD-PAP, Randox Laboratories Ltd., London, UK) method; TG and HDL-C were measured by cholesterol oxidase-phenol and amino phenazone (CHOD-PAP, Kyowa Medex Co., Ltd., Tokyo, Japan) method in a national lab in Beijing in 2015 and 2018.

#### 2.2.4. Other Variables

The demographic data, lifestyle, dietary information and other information involved in this study were obtained through face-to-face surveys with special questionnaires by uniformly trained and qualified investigators. The age groups were divided into three groups (18–49 years, 50–64 years, 65 years and above). The annual household income per capita was divided into three groups according to the quintile: low, middle and high. The education level was divided into three groups: low (primary school and below), middle (junior and senior high school) and high (college and above). Residence was divided into urban and rural areas. Current smoking and alcohol consumption in the past year were divided into yes and no groups. Physical activity levels included leisure physical activity, traffic physical activity, occupational physical activity and household physical activity; this was assessed using the product of the metabolic equivalent (MET) of each activity and the number of hours per week spent in various physical activities (hours/week) [23,24] and was divided into three groups according to the quartile: low, medium and high. Baseline energy, dietary fiber and dietary calcium were adjusted for continuous variables. Body Mass Index (BMI) is calculated by weight divided by height squared and is defined as <18.5 kg/m^2^, 18.5~23.9 kg/m^2^ and ≥24.0 kg/m^2^.

### 2.3. Statistical Analysis

Quantitative and qualitative variables were described by mean, standard deviation and percentage (%), respectively. Chi-square test and ANOVA were used to analyze the demographic characteristics of subjects with different dietary magnesium intake levels. Dietary magnesium intake was divided into quintile groups, and the risk association between dietary magnesium and MetS and its single component was analyzed by a multivariate Cox proportional risk model. The median values of dietary magnesium levels were inserted into the model as continuous variables for a trend test. Finally, a restricted cubic spline (RCS) model with 5 knots was used to analyze the dose-response relationship between dietary magnesium and MetS and its single component. All data were analyzed using SAS software package version 9.4 (SAS Institute, Inc., Cary, NC, USA) and R software version 4.1.0 (The R Foundation for Statistical Computing) and we defined statistical significance as *p* < 0.05. 

## 3. Results

### 3.1. Baseline Characteristics

The baseline characteristics of subjects divided by quintile of dietary magnesium intake were shown in Table 1. Compared with those in the top quintile of magnesium intake, those with lower dietary magnesium intake were more likely to have higher income, higher education, lower levels of physical activity, lower energy, dietary fiber and dietary calcium intake or to live in rural areas. Baseline waist circumference, HDL-C, systolic blood pressure, diastolic blood pressure and fasting blood glucose were different between groups (*p* < 0.05). Other variables, including age, sex, BMI, smoking, alcohol consumption and baseline TG levels, were not significantly different between groups with different dietary magnesium intake levels (*p* > 0.05).

### 3.2. Multivariate Cox Proportional Risk Regression Analysis of Dietary Magnesium Intake to MetS and Its Components

The study included 6104 participants who did not have MetS at baseline; 2024 developed the disease during a mean follow-up of 6.09 years, with an incidence of 33.16%. After adjustment for demographic characteristics, lifestyle and dietary factors, the risk of MetS was reduced by 16% in the highest quintile with dietary magnesium intake (HR = 0.84, 95% CI = 0.71–0.99, *p* trend = 0.32), using the lowest quintile as a reference.

The single-component study of MetS found that after adjusting for all covariates, the risk of central obesity, elevated TG, elevated blood pressure and elevated blood glucose was reduced by 18% (HR = 0.82, 95% CI = 0.69–0.98, *p* trend <0.05), 41% (HR = 0.59, 95% CI = 0.51–0.70, *p* trend < 0.05), 20% (HR = 0.80, 95% CI = 0.69–0.94, *p* trend = 0.08) and 42% (HR = 0.58, 95% CI = 0.50–0.67, *p* trend < 0.05). Decreased HDL-C was associated with a 23% lower risk in the third quintile of dietary magnesium intake (HR = 0.77, 95% CI = 0.65–0.93) and an increased risk in the highest quintile (HR = 1.21, 95% CI = 1.00–1.47) (Table 2).

### 3.3. Dose-Response Relationship between Dietary Magnesium Intake and MetS and Its Components

RCS analysis showed (Figure 2) that the fifth percentile of dietary magnesium intake was used as a reference, the overall and non-linear associations between dietary magnesium and the risk of MetS were statistically significant (*p* < 0.05), and the intake below 280 mg/day significantly lowered the risk for MetS. When dietary intake of magnesium was greater than 280 mg/day, the curve leveled off, but there was still a protective effect. There was no statistical association between dietary magnesium intake and MetS when the intake exceeded 500 mg/day.

The single-component study found that the overall u-shaped relationship between central obesity and decreased HDL-C and dietary magnesium intake was observed. The cut-off point was 280 mg/day. When the intake was lower than 280 mg/day, the risk of central obesity and decreased HDL-C decreased significantly, and then the curve showed an upward trend; dietary magnesium had no protective effect after 450 mg/day and 320 mg/day, respectively. In addition, the linear relationship between elevated TG, elevated blood pressure and elevated blood glucose and dietary magnesium was close, and the risk of disease was significantly reduced before magnesium intake was 280 mg/day, and the curve tended to be flat after 280 mg/day. All single components were non-linear association with MetS (*p* < 0.05).

## 4. Discussion

This study analyzed the relationship between dietary magnesium intake and MetS in adults aged 18 years and above in 15 provinces of China by the longitudinal data from the CHNS. The results showed that in a certain range, dietary magnesium was negatively and non-linearly correlated with MetS and its single component.

In a 15-year follow-up of 4637 adults aged 18–30 in the United States, He et al. found that the risk of MetS was reduced by 31% (HR = 0.69, 95%CI =0.52–0.91) in the highest quartile of dietary magnesium intake [25] (Adult treatment panel III, ATPIII). In two other studies of Arab and Italian adults, in the lowest quartile of dietary magnesium intake, the risk of MetS increased by 2.7 times (OR = 2.7, 95%CI = 1.0–7.2) [26] (International Diabetes Federation, IDF) and 3.26 times (OR = 3.26, 95%CI = 2.41–4.41) [27] (ATPIII), but the association between dietary magnesium and MetS disappeared in Italian adults after adjustment for dietary fiber. In this study, magnesium intake was highly correlated with fiber intake (r = 0.81), which may be caused by the similarity of foods providing fiber and magnesium. A similar situation occurred in the study of the Iranian population [28]. Therefore, multicollinearity caused by the high correlation between nutrients is worth considering. In our study, dietary fiber adjustment did not affect the association between dietary magnesium and MetS. In South Korea’s National Health and Nutrition Survey, an association between dietary magnesium and MetS was found only in women [14]. Another study of kidney transplant patients followed up for one year found that dietary magnesium in the highest quintile group had no statistical association with the risk of MetS (OR = 0.80, 95% CI = 0.16–3.86) [15], which was thought to be related to the selected group (non-healthy group) and short follow-up time, and no conclusions can be drawn about the long-term effects of dietary magnesium on MetS.

There are several explanations for the association between dietary magnesium and MetS. (1) Insulin resistance partially mediates the association between them, and a recent study was the first to quantify the mediating role of insulin resistance; the calculated percentage of mediation was 19.6%. Dietary magnesium exerts insulin effects by activating β subunits of insulin receptors and substrates and proteins in insulin signaling pathways [29]. (2) Relevant studies have also emphasized the core role of inflammation and oxidative stress in MetS [30]. (3) Low magnesium diet could result in serum and intracellular Mg^2+^ deficiency, which was especially evident in obese people with MetS [31]. (4) The role of dietary magnesium was also affected by Ca^2+^, and participants who met the recommended daily intakes of Mg^2+^ and Ca^2+^ had a reduced risk of MetS [31]. (5) In addition to the mechanisms above, genes also play an important role in the development of MetS [32].

In this study, higher dietary magnesium intake reduced the risk of central obesity (18%), elevated TG (41%), decreased HDL-C (23%), elevated blood pressure (20%) and elevated blood glucose (42%). Another cross-sectional data analysis of CHNS2009 also found the protective effect of dietary magnesium on the five components of MetS [29]. In other related studies, dietary magnesium intake was only significantly associated with individual components of MetS [15,25,28]. Previous studies had reported the protective effect of dietary magnesium on markers of obesity (BMI or waist circumference). In Mexican adults, an increase in daily magnesium intake of 10 mg/1000 kcal is associated with an average decrease in waist circumference of 0.49 cm (95% CI = −0.92, −0.07) [33] because magnesium reduces the absorption of fatty acids and cholesterol by forming difficult-to-absorb fatty acid salts in the gut, thus reducing energy intake in the diet and preventing the accumulation of adipose tissue [11]. Magnesium also acts as a co-factor of several key enzymes in lipid metabolism, improving HDL-C levels and reducing TG levels by affecting related enzyme activities [11]. However, no association between dietary magnesium and increased TG was found in two studies of American adults [5,25]. Low magnesium intake is an independent predictor of hypertension. Magnesium can change the permeability of cell membranes to calcium and sodium, which is also an important mechanism for the development of hypertension. In addition, magnesium can act as a calcium antagonist and regulate the tension and contractility of vascular smooth muscle by affecting calcium ion concentration, thus causing vascular dilation [34]. In addition, meta-analysis showed that dietary magnesium was significantly negatively correlated with diabetes [35]. Magnesium can play a role in insulin secretion by influencing calcium homeostasis and oxidative stress and can also act as a co-factor to regulate glucose metabolism [11]. In conclusion, dietary magnesium has a certain protective effect on the five components of MetS. However, the dose-response relationship of single components of MetS is different, especially at high dietary magnesium intake, so the specific mechanism needs to be further explored.

At present, there are few studies on the dose-response relationship between dietary magnesium and MetS and its components. A meta-analysis by Ju et al. found a linear relationship between dietary magnesium and MetS, with an increase of 150 mg/day, the risk of MetS decreased by 12% (RR = 0.88, 95% CI = 0.84–0.93) [12]. Han et al. [34] and Xu et al. [35] found a non-linear relationship between dietary magnesium and hypertension and diabetes, respectively, and believed that 300 mg/day was the best dose to prevent diabetes.

There is still no conclusive evidence of beneficial doses of dietary magnesium. Balance data obtained since 1997 showed that the recommended dietary allowance for magnesium is 250 mg/day for a healthy individual weighing 70 kg, and based on weight gain or loss, the standard magnesium requirement is less than 200 mg/day for German women and less than 250 mg/day for German men [36]. In most studies on the association between dietary magnesium and chronic diseases such as cardiovascular disease, hypertension and MetS, the difference was most significant when the average minimum intake was less than 250 mg/day [37]. In our study, magnesium intake of 280 mg/day was the best reference dose to prevent MetS and its single component and showed a non-linear relationship with both. It could provide a reference for the recommended intake of chronic diseases.

In this study, three years of follow-up data from CHNS were selected to explore the association between dietary magnesium and MetS and its single component based on large-scale population data, and the dose-response relationship was further analyzed. There are still some limitations in this study. First, the 3 d-24 h dietary recalls used to calculate magnesium intake are considered to have similar accuracy to the semi-quantitative food frequency questionnaire but have the disadvantage of recall bias. Second, simultaneous adjustment of highly correlated nutrients may not be the best method to independently identify the effects of a single nutrient because, with the increase of col-linearity, regression coefficients and their standard errors tend to be unstable, making interpretation of the results difficult. Finally, we cannot rule out the possibility of unknown confounding factors.

## 5. Conclusions

This study provides prospective evidence that dietary magnesium intake is negatively correlated with MetS and its single component and showed a significant non-linear correlation. When dietary magnesium intake was less than 280 mg/day, the risk of MetS and its single component decreased significantly with the increase in dietary magnesium intake, but the non-linear trend of the different components was slightly different. Therefore, the mechanism of dietary magnesium on a single component needs to be explored further in the future. In addition, the dietary magnesium intake of Chinese residents is at a low level and shows a downward trend. Therefore, it is recommended that residents consume more foods rich in magnesium and emphasize the improvement of overall dietary quality in order to reduce the risk of MetS effectively.

## Figures and Tables

**Figure 1 nutrients-14-02013-f001:**
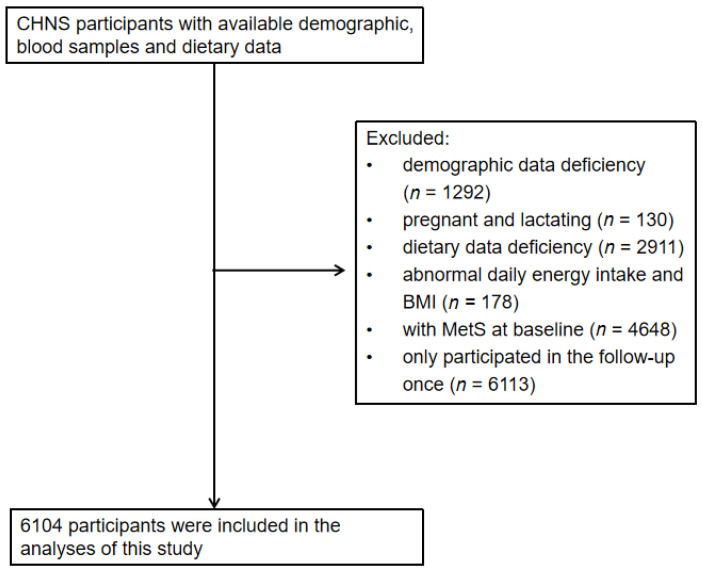
Flowchart of subjects.

**Figure 2 nutrients-14-02013-f002:**
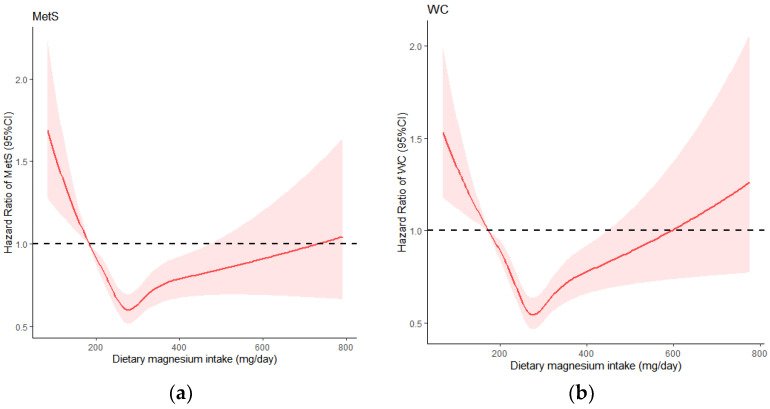
Dose-response relationship of dietary magnesium with MetS and its components: (**a**) MetS: metabolic syndrome; (**b**) WC: waist circumference; (**c**) TG: triglyceride; (**d**) HDL-C: high density lipoprotein cholesterol; (**e**) BP: blood pressure; (**f**) FPG: fasting plasma glucose.

**Table 1 nutrients-14-02013-t001:** Baseline characteristics according to quintile of dietary magnesium intake.

	Quintile of Dietary Magnesium Intake (mg/day)					
	Q1 (<225.89 mg/day)	Q2 (225.89~256.12 mg/day)	Q3 (256.12~287.23 mg/day)	Q4 (287.23~329.91 mg/day)	Q5 (≥329.91 mg/day)	*p*-Value
Age, %						0.22
18~49	51.31	53.48	51.92	48.98	48.81	
50~64	36.15	33.82	33.58	36.61	37.10	
65~	12.54	12.69	14.50	14.41	14.09	
Male, %	49.51	44.64	45.37	45.45	46.36	0.13
Household incomeper capita, %						<0.05
Low	27.62	31.20	34.97	36.20	36.69	
Median	31.72	30.30	35.30	35.22	34.07	
High	40.66	38.49	29.73	28.58	29.24	
Education, %						<0.05
Primary and below	33.61	32.19	36.94	37.18	40.13	
Middle and high	47.62	46.19	46.6	46.93	45.21	
College and above	18.77	21.62	16.46	15.89	14.66	
Urban, %	30.49	33.82	37.26	34.73	34.97	<0.05
Never smoked, %	70.82	71.25	70.68	69.7	71.01	0.93
Never drunk alcohol, %	68.77	67.73	68.39	68.39	67.98	0.99
Physical activity, %						<0.05
Low	35.90	32.51	33.82	32.76	31.61	
Median	35.33	36.45	33.01	31.37	30.55	
High	28.77	31.04	33.17	35.87	37.84	
BMI (kg/m^2^), %						0.08
<18.5	6.72	7.21	6.63	7.37	5.16	
18.5~23.9	62.46	60.85	62.65	59.79	59.05	
≥24.0	30.82	31.94	30.71	32.84	35.79	
Energy(kcal/day)	2201.10 ± 795.07	2049.50 ± 658.96	2014.17 ± 611.07	2043.68 ± 625.45	2335.93 ± 744.84	<0.05
Dietary fibre (mg/day)	9.02 ± 4.68	9.86 ± 5.15	10.83 ± 5.51	12.34 ± 5.67	18.44 ± 11.58	<0.05
Calcium (mg/day)	295.52 ± 134.24	325.25 ± 142.77	368.85 ± 221.44	406.62 ± 167.17	554.98 ± 334.05	<0.05
WC (cm)	79.69 ± 10.35	79.69 ± 9.66	79.85 ± 9.96	80.66 ± 9.53	80.83 ± 11.62	<0.05
TG (mmol/L)	1.19 ± 0.70	1.19 ± 0.73	1.20 ± 0.73	1.19 ± 0.83	1.21 ± 0.90	0.99
HDL-C(mmol/L)	1.44 ± 0.34	1.47 ± 0.36	1.50 ± 0.38	1.50 ± 0.50	1.49 ± 0.40	<0.05
SBP (mmHg)	120.43 ± 15.95	120.47 ± 16.43	121.58 ± 16.40	121.21 ± 16.27	122.47 ± 15.00	<0.05
DBP (mmHg)	77.43 ± 9,85	77.53 ± 9,84	78.69 ± 9.92	78.29 ± 10.19	79.32 ± 9.42	<0.05
FPG (mmol/L)	5.09 ± 0.87	5.13 ± 1.07	5.10 ± 0.87	5.02 ± 0.84	5.03 ± 0.97	<0.05

Values are mean ± SD for continuous variables and percentage for categorical variables. WC: waist circumference; TG: triglyceride; HDL-C: high density lipoprotein cholesterol; SBP: systolic blood pressure; DBP: diastolic blood pressure; FPG: fasting plasma glucose.

**Table 2 nutrients-14-02013-t002:** Multivariate Cox proportional risk regression analysis of dietary magnesium intake to MetS and its components [HR (95% CI)].

	Quintile of Dietary Magnesium Intake (mg/day)					
	Q1	Q2	Q3	Q4	Q5	*p* trend
	Metabolic syndrome ^a^					
Median (mg/day)	202.82	242.35	271.36	305.33	370.72	
Model 1	1.00 (ref)	0.77 (0.67, 0.90) *	0.74 (0.64, 0.85) *	0.81 (0.70, 0.93) *	0.91 (0.79, 1.04)	0.96
Model 2	1.00 (ref)	0.79 (0.68, 0.91) *	0.76 (0.65, 0.88) *	0.84 (0.72, 0.96) *	0.95 (0.82,1.09)	0.60
Model 3	1.00 (ref)	0.76 (0.65, 0.88) *	0.72 (0.62, 0.82) *	0.79 (0.68, 0.92) *	0.92 (0.78,1.09)	0.86
Model 4	1.00 (ref)	0.75 (0.64, 0.88) *	0.69 (0.59, 0.80) *	0.76 (0.65, 0.88) *	0.84 (0.71, 0.99) *	0.32
	Central obesity ^b^					
Median (mg/day)	195.81	238.45	268.15	304.03	369.22	
Model 1	1.00 (ref)	0.80 (0.69, 0.93) *	0.65 (0.56, 0.75) *	0.70 (0.61, 0.81) *	0.89 (0.77,1.02)	0.13
Model 2	1.00 (ref)	0.80 (0.69, 0.93) *	0.66 (0.57, 0.77) *	0.70 (0.60, 0.81) *	0.89 (0.77,1.02)	0.12
Model 3	1.00 (ref)	0.77 (0.66, 0.90) *	0.63 (0.54, 0.74) *	0.66 (0.57, 0.78) *	0.88 (0.74,1.04)	0.07
Model 4	1.00 (ref)	0.80 (0.68, 0.94) *	0.64 (0.54, 0.75) *	0.65 (0.55, 0.77) *	0.82 (0.69, 0.98) *	<0.05
	Elevated TG ^c^					
Median (mg/day)	194.14	236.18	265.51	300.72	365.54	
Model 1	1.00 (ref)	0.70 (0.61, 0.80) *	0.57 (0.50, 0.65) *	0.57 (0.50, 0.65) *	0.66 (0.58, 0.75) *	<0.05
Model 2	1.00 (ref)	0.71 (0.62, 0.81) *	0.59 (0.51, 0.67) *	0.58 (0.50, 0.66) *	0.67 (0.59, 0.77) *	<0.05
Model 3	1.00 (ref)	0.66 (0.58, 0.75) *	0.54 (0.47, 0.62) *	0.52 (0.45, 0.59) *	0.60 (0.51, 0.70) *	<0.05
Model 4	1.00 (ref)	0.67 (0.59, 0.77) *	0.56 (0.49, 0.65) *	0.52 (0.45, 0.60) *	0.59 (0.51, 0.70) *	<0.05
	Decreased HDL-C ^d^					
Median (mg/day)	202.89	243.64	273.21	307.60	371.13	
Model 1	1.00 (ref)	0.93 (0.79,1.10)	0.78 (0.66, 0.92) *	0.97 (0.82,1.13)	1.11 (0.95,1.30)	<0.05
Model 2	1.00 (ref)	0.94 (0.79,1.11)	0.78 (0.66, 0.92) *	0.96 (0.82,1.13)	1.12 (0.95,1.31)	<0.05
Model 3	1.00 (ref)	0.93 (0.78,1.10)	0.77 (0.65, 0.92) *	0.98 (0.82,1.16)	1.23(1.03,1.48) *	<0.05
Model 4	1.00 (ref)	0.91 (0.76,1.09)	0.77 (0.65, 0.93) *	0.96 (0.80,1.14)	1.21(1.00,1.47) *	<0.05
	Elevated BP ^e^					
Median (mg/day)	196.72	238.93	269.01	304.21	365.07	
Model 1	1.00 (ref)	0.73(0,64, 0.84) *	0.71 (0.62, 0.81) *	0.72 (0.64, 0.82) *	0.83 (0.73, 0.94) *	0.10
Model 2	1.00 (ref)	0.74 (0.64, 0.84) *	0.71 (0.62, 0.81) *	0.72 (0.63, 0.82) *	0.83 (0.73, 0.94) *	0.10
Model 3	1.00 (ref)	0.71 (0.62, 0.82) *	0.68 (0.59, 0.78) *	0.69 (0.60, 0.79) *	0.83 (0.71, 0.96) *	0.16
Model 4	1.00 (ref)	0.72 (0.63, 0.83) *	0.67 (0.58, 0.77) *	0.70 (0.61, 0.81) *	0.80 (0.69, 0.94) *	0.08
	Elevated FPG ^f^					
Median (mg/day)	195.46	237.58	267.21	303.01	366.81	
Model 1	1.00 (ref)	0.71 (0.63, 0.80) *	0.56 (0.50, 0.63) *	0.57 (0.51, 0.64) *	0.63 (0.56, 0.70) *	<0.05
Model 2	1.00 (ref)	0.73 (0.64, 0.82) *	0.57 (0.51, 0.65) *	0.58 (0.51, 0.65) *	0.64 (0.57, 0.72) *	<0.05
Model 3	1.00 (ref)	0.68 (0.60, 0.77) *	0.53 (0.47, 0.60) *	0.52 (0.46, 0.60) *	0.60 (0.52, 0.69) *	<0.05
Model 4	1.00 (ref)	0.68 (0.60, 0.77) *	0.53 (0.46, 0.60) *	0.51 (0.45, 0.58) *	0.58 (0.50, 0.67) *	<0.05

* *p <* 0.05. ^a^
*n* = 6104. ^b^ *n* = 4976. ^c^ *n* = 7918. ^d^
*n* = 5430. ^e^ *n* = 5768. ^f^
*n* = 7977. Model 1 adjusted age, sex, education, urban and rural areas, and income; In model 2, smoking, drinking and physical activity levels were further adjusted based on Model 1. In Model 3, energy, dietary fiber and calcium were further adjusted based on Model 2. Model 4 further adjusted BMI on the basis of Model 3.

## Data Availability

Data sharing is not applicable to this article.

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
