# Peer review of "Relationship between Dietary Magnesium Intake and Metabolic Syndrome"

_nutrients, 2022, doi:10.3390/nu14102013_

Round 1

Reviewer 1 Report

Dear authors, I congratulate them on the very interesting work, particularly for the dose-response curve of magnesium and components of MetS.

Here are my suggestions and advice for completing the work:

  1. lines 23-25 this sentence has no sense. The components of MetS were reduced by Magnesium intake?
  2. section 2.1 a graph of excluded/included subjects must be inserted.
  3. lines 86-88 better explain this sentence, the sense is not clear.
  4. line 150 not quartile but quintile.
  5. line 216 a typo? (ATPIII etc.) and (IDF )
  6. line 233 the authors explain better with more references to the mechanism of action between magnesium and Mets, there are many papers about this topic.
    e.g. Piuri, G.; Zocchi, M.; Della Porta, M.; Ficara, V.; Manoni, M.; Zuccotti, G.V.; Pinotti, L.; Maier, J.A.; Cazzola, R. Magnesium in Obesity, Metabolic Syndrome, and Type 2 Diabetes. Nutrients 2021, 13, 320. https://doi.org/10.3390/nu13020320
  7. Line 248-250 this sentence is not supported by reference 34. Find a reference that supports this sentence or delete it.
  8. The conclusion must be improved, and highlight the results of the dose-response curve.

Author Response

Thanks for your hard review, the specific modifications are as follows :

Point 1:  lines 23-25 this sentence has no sense. The components of MetS were reduced by Magnesium intake?

Response 1: Modify this sentence with the following description:

Central obesity, elevated TG, elevated blood pressure, and elevated blood glucose was reduced by 18%, 41%, 20% , and 42%, respectively. The risk of decreased HDL-C was reduced by 23% in the third quintile of dietary magnesium intake, with a slightly increased risk in the highest group.  

Point 2: section 2.1 a graph of excluded/included subjects must be inserted.

Response 2: a graph of excluded/included subjects has been inserted in section2.

Point 3:  lines 86-88 better explain this sentence, the sense is not clear.

Response 3: In prospective analyses, considering that potential changes in diet after the development of disease may confound the relationship between dietary magnesium intake and MetS and its components, updating of dietary information was stopped upon diagnosis of the disease.

Point 4:  line 150 not quartile but quintile.

Response 4:  “Quartile” in line 150 has been changed “quintile”

Point 5:  line 216 a typo? (ATPIII etc.) and (IDF )

Response 5:  ATPIII (Adult treatment panel III) and IDF (International Diabetes Federation) are different diagnostic criteria for MetS. Full names and abbreviations have been added to the manuscript.

Point 6: line 233 the authors explain better with more references to the mechanism of action between magnesium and Mets, there are many papers about this topic.
e.g. Piuri, G.; Zocchi, M.; Della Porta, M.; Ficara, V.; Manoni, M.; Zuccotti, G.V.; Pinotti, L.; Maier, J.A.; Cazzola, R. Magnesium in Obesity, Metabolic Syndrome, and Type 2 Diabetes. Nutrients 2021, 13, 320. https://doi.org/10.3390/nu13020320

Response 6: There are several explanations for the association between dietary magnesium and MetS.

(1) Insulin resistance partially mediates the association between them, and a recent study was the first to quantify the mediating role of insulin resistance, and the calculated percentage of mediation was 19.6%. Dietary magnesium exerts insulin effects by activating β subunits of insulin receptors and substrates and proteins in insulin signaling pathways. (2) Relevant studies have also emphasized the core role of inflammation and oxidative stress in MetS . (3) Low magnesium diet results in serum and intracellular Mg2+ deficiency. This was especially evident in obese people with MetS [31]. (4) The role of dietary magnesium is also affected by Ca2+, and participants who meet the recommended daily intakes of Mg2+ and Ca2+ had a reduced risk of MetS . (5) In addition to the mechanisms above, genes also play an important role in the development of MetS .

Point 7:  Line 248-250 this sentence is not supported by reference 34. Find a reference that supports this sentence or delete it.

Response 7: Relevant literature supporting this sentence has been found and inserted.

Point 8:   The conclusion must be improved, and highlight the results of the dose-response curve.

Response 8: Combined with the results of the dose-response relationship between dietary magnesium and MetS, the conclusion has been improved.

This study provides prospective evidence that dietary magnesium intake is negatively correlated with MetS and its single component and showed a significant non-linear correlation. When dietary magnesium intake was less than 280mg/d, the risk of MetS and its single component decreased significantly with the increase of dietary magnesium intake, but the non-linear trend of different component was slightly different. Therefore, the mechanism of dietary magnesium on single component needs to be explored in the future. In addition, the dietary magnesium intake of Chinese residents is at a low level and showing a downward trend.Therefore, it is recommended that residents consume more foods rich in magnesium and emphasize the improvement of overall dietary quality in order to reduce the risk of MetS effectively.

Reviewer 2 Report

In this prospective study, the authors provide evidence that a correlation exists between magnesium level and onset of metabolic syndrome. The data have been obtained in a population sample study sufficiently large to support the conclusion.

Comments:

1. The authors evidenced that a level of 280 mg is the cut off at which most of the modifications induced by or observed with the metabolic syndrome occur.  It is unclear from the data presented in Fig. 1 whether the authors attempted to restore Mg2+ or whether the data point at 320 mg/d or 450 mg/d were extrapolation based on the obtained data. Also, from the data reported in Fig 1, it would appear that the concentration of Mg2+ should be around 800 mg/d for te hazard ration of MS and WC to reach level 1, and this dose will still not restore the TG. What is the interpretation the authors offer for this discrepancy? 

2. Also, it is the understanding of this reviewer that the Mg2+ level was measured in the serum and all the extrapolations were based on this parameter. As Mg2+ is primarily an intracellular cation, how this serum value correlates with cellular Mg2+ content? Do the authors have any data addressing this point? Would it not be possible that a modest increase in cellular Mg2+ could restore some of the measured parameters to normal?

Author Response

Point 1: The authors evidenced that a level of 280 mg is the cut off at which most of the modifications induced by or observed with the metabolic syndrome occur.  It is unclear from the data presented in Fig. 1 whether the authors attempted to restore Mg2+ or whether the data point at 320 mg/d or 450 mg/d were extrapolation based on the obtained data. Also, from the data reported in Fig 1, it would appear that the concentration of Mg2+ should be around 800 mg/d for the hazard ration of MS and WC to reach level 1, and this dose will still not restore the TG. What is the interpretation the authors offer for this discrepancy? 

Response 1: This study collected food consumption data of the population, calculated dietary magnesium intake according to the food composition table, and fitted the dose-response curve of dietary magnesium with MetS and its components. Therefore, 320mg/d and 450mg/d are derived from existing data rather than extrapolated. In addition, there were differences in the effects of dietary magnesium on individual components of MetS, especially at high intakes, which may be related to different mechanisms of magnesium. As a cofactor in more than 300 enzyme systems, magnesium regulates a variety of biochemical reactions in the body, including protein synthesis, muscle and nerve transmission, neuromuscular transmission, blood glucose control and blood pressure regulation, etc, and different sensitivities at different sites of action may influence the outcome. The specific mechanisms need to be further explored.

Point 2:  Also, it is the understanding of this reviewer that the Mg2+ level was measured in the serum and all the extrapolations were based on this parameter. As Mg2+ is primarily an intracellular cation, how this serum value correlates with cellular Mg2+ content? Do the authors have any data addressing this point? Would it not be possible that a modest increase in cellular Mg2+ could restore some of the measured parameters to normal?

Response 2: Thank you for giving me valuable advises. The main purpose of this study was to study the relationship between dietary magnesium intake and MetS, and to find the cut-off value, so as to provide reference for the recommended intake of chronic diseases in the population. The association between serum and cellular magnesium mentioned in your recommendation, relevant studies found that low magnesium diet could lead to the deficiency of serum magnesium and cellular magnesium, which was more obvious in obese people with MetS. And some studies also  have found that the correlation between dietary magnesium and serum magnesium is very low (r=0.049). Due to the limitations of this study, there is no data to study the association between dietary magnesium, serum magnesium and cellular magnesium. It is hoped that special studies will be conducted in the future to explore this association and metabolic mechanism.  
